



# Statistical analyzing the effect of ionospheric irregularity
# on GNSS radio occultation atmospheric measurement
Mingzhe Li [1,2,3,4], Xinan Yue [1,2,3,4]
[1] Key Laboratory of Earth and Planetary Physics, Institute of Geology and Geophysics, Chinese Academy
of Sciences, Beijing, China
[2] Innovation Academy for Earth Science, CAS, Beijing, China
[3] Beijing National Observatory of Space Environment, Institute of Geology and Geophysics, Chinese
Academy of Sciences, Beijing, China
[4] College of Earth and Planetary Sciences, University of Chinese Academy of Sciences, Beijing, China
*Correspondence to*: Xinan Yue (yuexinan@mail.iggcas.ac.cn)
**Abstract**. The Global Navigation Satellite System (GNSS) atmospheric radio occultation (RO) has been
an effective method for Earth's atmosphere exploring. RO signals propagate through ionosphere before
reaching the neutral atmosphere. The GNSS signal is affected by the ionospheric irregularity including
the sporadic E (Es) and the F region irregularity due to mainly multipath effect. The effect of ionospheric
irregularity on atmospheric RO data has been demonstrated by several studies in terms of cases.
However, its statistical effect has not been investigated comprehensively. In this study, based on the
Constellation Observing System for Meteorology, Ionosphere, and Climate (COSMIC) RO data during
2011-2013, the failed inverted RO events occurrence rate and the bending angle oscillation, which is
defined as the standard deviation of the bias between the observed bending angle and the National Center
for Atmospheric Research (NCAR) climatology model bending angle between 60 and 80 km, were used
for statistical analysis. It is found that in middle and low latitudes during the daytime, the failed inverted
RO occurrence and the bending angle oscillation show obvious latitude, longitude, and local time
variations, which correspond well with the Es occurrence features. The F region irregularity (FI)
contributes to the obvious increase of the failed inverted RO occurrence rate and the bending angle
oscillation value during the nighttime over the geomagnetic equatorial regions. For high latitude regions,
the Es can increase the failed inverted RO occurrence rate and the bending angle oscillation value during
the nighttime. There also exists the seasonal dependency of the failed inverted RO event and the bending
angle oscillation. Overall, the ionospheric irregularity effects on GNSS atmospheric RO measurement



exist in terms of failed RO event inversion and bending angle oscillation statistically. Awareness of these
effects could benefit both the data retrieval and applications of RO in the lower atmosphere.

**1.  Introduction**
The radio occultation (RO) is a technique originally developed in the late 1960s and early 1970s for
planetary atmosphere exploring. With the great development of the Global Navigation Satellite System
(GNSS) over the past 30 years, the GNSS signal has been an effective source for exploring the Earth's
atmosphere. Several RO missions such as the Global Positioning System Meteorology (GPS/MET), the
Challenging Minisatellite Payload (CHAMP) (Wickert et al., 2001), the Scientific Application Satellite-
C (SAC-C), the Gravity Recovery and Climate Experiment (GRACE) (Beyerle et al., 2005), the
Constellation Observing System for Meteorology, Ionosphere and Climate (COSMIC) (Schreiner et al.,
2007), the Meteorological Operational Satellite Program (Metop) A/B, the Fengyun-3C (FY-3C) (Mao
et al., 2016) and et al have proven the good capability of RO for observing the Earth's ionosphere and
atmosphere. High-quality products of RO have been used for space weather, weather and climate
research (Anthes et al., 2008).

The RO technique could be divided into the ionospheric RO and the atmospheric RO. For the former one,
the GNSS signal propagates through the ionosphere. Dual-frequency pseudo-range and carrier phase can
be observed by the receiver onboard the low Earth orbit (LEO) satellite and be used to invert the electron
density profile. Besides, the amplitude and phase measurements can be used to calculate the ionospheric
scintillation index such as the S4 index, which can represent the occurrence of the ionospheric irregularity
(Yue et al., 2016). For the latter one, the GNSS signal propagates through both the ionosphere and the
neutral atmosphere. The dual-frequency carrier phase can be used to calculate the bending angle and then
invert the atmospheric parameters. As a result, the effects on signals caused by the ionosphere should be
removed before deriving the atmospheric RO products.

For atmospheric RO, the GNSS signal is mainly affected by the ionosphere in two ways. Firstly, the
existence of dense ionospheric electron density contributes to the bending of signals. Similar to the first-
order ionospheric term calibration used in ground-based dual-frequency observations, a linear





combination of the two-band signal bending at the same impact parameter is usually used to remove the
ionospheric effect (Vorob'ev and Krasil'nikova, 1994). However, after the linear combination of bending
angles, there still exists a residual ionospheric error (RIE). The RIE could bring ionospheric variability
such as solar cycle, local time, and seasonal variations into atmospheric RO products although its
amplitude is relatively low (Li et al., 2020). It means that the climate research using atmospheric RO
products would be affected by the ionosphere. Some efforts have been tried for the RIE calibration
(Danzer et al., 2015; Liu et al., 2018; Li et al., 2020). In our previous study, we have characterized this
effect statistically by both ray-tracing simulation and data analysis (Li et al., 2020). Secondly, the small-
scale irregularities in the ionosphere also have an impact on the GNSS signal and finally affect the
atmospheric RO products. The small-scale irregularities of which interest to this study are the sporadic
E (Es) and the F region irregularity (FI). As indicated by former studies, the ionospheric irregularity will
cause refraction or diffraction of the GNSS signal during its propagating through the ionosphere. The
received signals could show temporal fluctuations in both amplitude and phase, which is known as the
ionospheric scintillation. The impact of the small-scale irregularity on atmospheric RO can be significant
but show quite different climatologically characteristics in comparison with the large-scale ionospheric
effects (Li et al., 2020). Besides, it is difficult to model the ionospheric irregularity in a deterministic
fashion for simulation research (Mannucci, et al., 2011). Previous studies only pointed out this small
scale ionospheric effect in terms of cases (Zeng and Sokolovskiy, 2010). To our knowledge, there is no
comprehensive study giving statistical analysis between ionospheric irregularity and atmospheric RO
products, which is quite important to quantify this effect and therefore benefit atmospheric RO data
retrieval and application. This is the main objective of this study.

In the following sections, we try to study the ionospheric irregularity effects on GNSS atmospheric RO
measurement statistically. Based on previous related studies, the current study will make sense as
following advantages: (1) the correlation between failed inverted COSMIC RO events and the
ionospheric irregularity are analyzed; (2) morphology of the bending angle oscillation in the atmospheric
RO measurement are presented in comparison with the occurrence rate of both Es and FI; (3) the seasonal
dependency of failed inverted RO events and the bending angle oscillation is analyzed. We will describe
the COSMIC observation and the statistical method in Section 2 and Section 3, respectively. Then the



ionospheric irregularity effects on single RO cases will be shown in Section 4. The statistical results of
the failed inverted RO event and the bending angle oscillation will be depicted in comparison with the
ionospheric irregularity occurrence rate in Section 5. Finally, the conclusions and implications will be
presented in Section 6.

**2. RO Data Description**
COSMIC, as one of the most successful RO missions, was launched on 15 April 2006. The constellation
with six LEO satellites has contributed millions of profiles for space weather, weather, and climate
research in the past 14 years. Each COSMIC satellite has four separate antennas: two high gain
occultation antennas received GNSS signals with a 50 Hz sampling rate to explore the neutral atmosphere
from the top (~130 km) to bottom or reverse. The carrier phase modulated on signals can be used for
excess phase calculating and atmospheric parameters retrieving. The other two antennas are precise orbit
determination (POD) antennas with a 1 Hz sampling rate. The received signals are used for LEO orbiting,
ionosphere electron density, slant total electron content (TEC), and scintillation index calculating
(Schreiner et al., 2007, 2011). The COSMIC data are processed by the COSMIC Data Analysis and
Archive Center (CDAAC) of the University Corporation for Atmospheric Research (UCAR) and
available on the CDAAC website (http://cdaac-www.cosmic.ucar.edu/). In this study, the COSMIC RO
observations during 2011-2013 were used for analysis. The S4 scintillation index in auxiliary data file
(scnLv1) was used for the Es and FI occurrence rates calculation. The dry atmospheric profiles file
(atmPrf) were used for the failed inverted RO occurrence rate calculation and the atmospheric bending
angle oscillation morphology analysis. The specific calculating method will be introduced in the
following section.

**3. Analysis Method**
To study the ionospheric irregularity effects on RO, we focus on analyzing two parameters: the failed
inverted RO event occurrence rate and the bending angle oscillation defined as the mean standard
deviation of the bias between the observed bending angle and the NCAR climatology model bending
angle during the 60-80 km altitude interval. The failed inverted RO event means those events flunked
the quality control during profiles inversion in CDAAC. They are identified by the 'bad' attribute in the



atmPrf file whose values are equal to 1. The oscillation of atmospheric RO bending angle is also provided
by the atmPrf file. The S4 index contained in the scnLv1 file is used to represent the occurrence of the
ionospheric irregularity. The occurrence rate of S4 index larger than 0.3 is set to represent the occurrence
of ionospheric irregularity. It should be noted that we identify the occurrence altitude range between
50~600 km as the contribution of both Es and FI together. We first calculated the failed inverted RO
event occurrence rate in comparison with the ionospheric irregularity occurrence rate. After that, failed
inverted RO events of COSMIC during 2011-2013 were screened out to study the Es and FI effects on
the bending angle observation. Additionally, the ionPrf file of CDAAC is used for displaying the electron
density profiles in single cases.

**4.    Ionospheric Irregularity Effect on Single RO Cases**
To have a preliminary knowledge of the ionospheric irregularity effect on atmospheric RO, we firstly
show several typical single case examples. The results are shown in Figure 1. From top to bottom, each
row show results of a case. The three cases represent the RO event without ionospheric irregularity, the
RO event affected by the Es, and the event affected by the FI, respectively. From left to right columns,
the panels represent the inverted bending angles, the signal-to-noise ratio (SNR) of L1 C/A signal, the
related electron density profile, and the inverted dry temperature profile compared with the European
Centre for Medium-Range Weather Forecasts (ECMWF) results. Please note that the grey lines in the
rightmost panels denote the results of RO inverted temperature and the red lines represent those of
ECMWF. The purple lines in case 1 and case 3 are the temperature bias multiplied by 10 for convenient
comparison. The features of L1 C/A SNR profile and the electron density profile could identify whether
a RO event is affected by the ionospheric irregularity (Yue et al., 2015). As depicted in the second column,
the case 2 shows visible peaks in SNR fluctuation around 110 km, which correspond to the occurrence
altitude range of Es and could reflect the Es effects on this event. Besides, the electron density profile of
case 3 shows obvious scintillation between 200 km and 300 km, which implies the FI impact on signals.
We also use the corresponding CDAAC scnLv1 file for verification. The S4 maximum values of cases
1-3 are 0.03, 0.59, and 1.21, with S4 peaks around 111.69, 108.45, and 274.86 km, respectively. It can
be seen that the normal case shows good inversion results. The value of the LC bending angle above 40
km is much smaller than those of the L1 and L2 bending angles. It means that the ionosphere dominates





the bending of the RO signal in this tangent altitude interval and the linear combination method works
well. The bias between the observed dry temperature and the ECMWF result is insignificant. However,
the Es case shows bad results. Values of the inverted L2 bending angle are negative, which leads to larger
values of LC bending angle after the linear combination. As a result, the temperature profile is failed
inverted. Significant temperature bias between the observation and model result can be seen from the
rightmost panel. For the FI case, oscillations can be seen in the LC bending angle profile in the leftmost
panel as well as the temperature bias profile in the rightmost panel. The bending angle oscillation values
of cases 1-3 are 0.66, 16.06, and 5.74 µrad, respectively. It indicates that this oscillation could also be
related to the ionospheric irregularity. The geometry of atmospheric RO observation determines that the
ionospheric irregularity effects could propagate to a deep tangent height, where far below the altitude
range of irregularity occurrence (Wu, 2020). We have gone through many cases and found that the failed
inverted RO event and strong bending angle oscillation occurs usually along with Es and FI occurrence.
But not all events affected by the Es and FI are failed inverted. The analysis of single cases motivates
our further statistical study.

**5.   Statistical Results**
The Es and FI have been investigated comprehensively in the past several decades (Hocke et al., 2001;
Straus et al., 2003; Wu, 2005; Arras et al., 2008, 2009; Carter et al., 2013; Yue et al., 2015, 2016).
Generally, the Es can be seen as thin layers with much higher plasma density than the normal E region
density occurring during the altitude range of ~ 90-120 km. The occurrence rate of Es is controlled by
many factors such as the tidal wind, the Earth's geomagnetic field, and metal ions (Axford, 1963; Chu
et al., 2014). These factors lead to the complicate variations of Es along with latitude, longitude, altitude,
local time, and season (Hocke et al., 2001; Wu, 2005; Arras et al., 2008, 2009). FI is the plasma
irregularity and inhomogeneity in the F region caused by plasma instabilities (Dungey, 1956; Fejer and
Kelley, 1980). The scale sizes of the density irregularity range from a few centimeters to hundreds of
kilometers and the irregularity can appear at all latitudes. Both Es and FI have been observed by
ionosonde, incoherent/coherent scatter radars, and ground-based GNSS network. Since the success of
GPS/MET, the GNSS RO has also been proven as an effective technique to detect the occurrence of Es
and FI. Hocke et al. (2001) first derived the occurrence of Es from the GPS/MET observation and



confirmed its seasonal variation. Wu (2005) studied the latitude, local time, altitude, and seasonal
dependency of Es by using CHAMP occultation data. Arras et al. (2008) further investigated the
occurrence of Es using multiple RO missions including CHAMP, GRACE-A, and COSMIC. Chu et al.
(2014) presented the morphology of Es based on COSMIC amplitude and phase fluctuations of L-band
signals. For FI, Straus et al. (2003) made a statistical analysis of the GPS C/A code SNR fluctuations on
L1 frequency based on observations onboard the PICOSat satellite. They found that the geographic and
local time distributions of occultation having large values of the S4 index were consistent with known
scintillation climatology. Brahmanandam et al. (2012) presented the three-dimensional global
morphology and seasonal variations of S4 index measured from COSMIC for a low solar activity year
2008 and found the latitude, altitude, and local time dependency of FI. Carter et al. (2013) further
revealed the longitudinal and seasonal variations of equatorial FI using COSMIC S4 index. Besides, Yue
et al. (2015, 2016) also studied the complex Es and the ionospheric irregularity related GPS RO loss of
lock by COSMIC S4 index.

As investigated in the single case section, the failed inverted RO event and bending angle oscillation
could be related to the ionospheric irregularity. So we carried out statistical research from all COSMIC
atmospheric events during 2011-2013. Firstly, the RO event whose 'bad' attribute in the atmPrf file equals
to 1 was selected as the failed inverted RO for the statistics. Then the failed inverted RO events were
screened out for the bending angle oscillation study. Both the geographical and geomagnetic distributions
of the ionospheric irregularity, the failed inverted RO event occurrence rate and the bending angle
oscillation are shown in Figure 2. The grid resolutions are 10°×3° for Lon×Lat and 3°×2 h for
MLat×MLT, respectively.

The global geographical distribution of both Es and FI occurrence rate together is depicted in the top left
panel. For low and middle latitudes, two peaks of Es occurrence rate locate in the East Asia region and
the North Africa region in the northern hemisphere. One peak locates near the South America region.
The values of occurrence rate are greater than 30% in peak regions. One trough can be seen around the
South Africa region with an occurrence rate lower than 10%. The result corresponds well with the Es
characteristics derived from the GPS RO phase and SNR fluctuations (Wu, 2005). Besides, an occurrence



enhancement can be seen around the West Africa and the Atlantic Ocean region, which agrees with the
previous studies base on COSMIC S4 index (Brahmanandam et al., 2012; Yue et al., 2016) and indicates
the contributions of FI. For high latitudes, two peaks are available during 120° W-150° W and 0°-60° E
in the northern hemisphere and one peak can be seen around 120° E near North Antarctica. In the top
middle panel, we plotted the occurrence rate of failed inverted RO events during 2011-2013. The rate
represents the failed inverted RO event in percentage which was calculated based on all observed
COSMIC RO events during this time interval. Overall, the global distribution of the failed inverted RO
event occurrence agrees with those of the Es occurrence in the top left panel. Two peaks in the northern
hemisphere and one in the southern hemisphere match the locations of Es occurrence peaks around ±20°.
Besides, there exists an obvious increase in the failed inverted RO events in high latitudes. It should be
noted that the occurrence rate distribution of failed inverted RO event can't match those of Es and FI
completely because the inversion error is not only affected by the ionospheric irregularity but also
affected by other factors such as the low SNR. However, the contribution of FI on the failed inverted RO
event is not obvious in this panel. This might be due to the high occurrence regions of FI overlaps those
of Es partly. In the top right panel, we plotted the global distributions of the median bending angle
oscillation. The results are also in good agreement with those patterns of Es and FI occurrence rate.
Strong oscillations can be seen around North Africa and the East Asia regions in the Northern Hemisphere
and around South America in the Southern Hemisphere, with bending angle oscillation values of ~1.4
μrad. The trough of bending angle oscillation can be seen around the South Africa regions with values
less than 1 μrad. Both the locations of peaks and troughs correspond well with those of Es occurrence
rate. Besides, larger oscillation values are available in the Atlantic Ocean around the equator, which could
be related to the high FI occurrence in these regions. Especially, both the failed inverted RO and the
bending angle oscillation show obvious peaks around 120° E near North Antarctica. Peaks of the two
parameters could be related to the high occurrence of both Es and FI in this region. Peaks of Es and FI
occurrence in this region have also been observed by Wu (2020) based on S4 index from RO data sets.

We also plotted the geomagnetic local time and latitude (MLT-MLat) distribution of the three parameters
in the bottom panels in Figure 2 for further comparison. In most regions of the bottom left panel, the
distributions are similar to those of the Es. Irregularity occurs more around geomagnetic equator regions



and the aurora oval regions. Around the geomagnetic equator regions during 18-24 MLT, there is an
occurrence enhancement caused by FI. Both Es and FI contributes to a 'three peaks' feature in the equator
regions after sunset. These features correspond to the previous studies observed no matter by ground-
based GNSS observations (Li et al., 2011) or COSMIC RO observations (Chen and Huang, 2017).
Similar features can be seen in the bottom middle panel. The occurrence rate of the failed inverted RO
event is higher around the equator regions from sunset to midnight, which can reach 20%. In high
latitudes, two ovals are available. Besides, the failed inverted RO event also occurs more after midnight
and during the noon in the Southern Hemisphere. In these regions, the FI could make contributions. For
bending angle oscillation in the bottom right panel, three peaks of the mean oscillation value exist along
the geomagnetic latitude, which denotes the contribution of Es and FI during the nighttime. The value of
bending angle oscillation tends to be small within $\pm 60°$ during 2~10 MLT. For high latitudes, the
bending angle shows strong oscillation even though the occurrence rate of Es and FI is lower than those
of peak regions in middle and low latitudes.

For a better display of the high-latitude results, we plotted the irregularity and failed inverted RO
occurrence rates as well as the bending angle oscillation variation with MLT-MLat in northern and
southern polar regions in Figure 3. As depicted in the left two panels, the ionospheric irregularity mainly
occurs during 18-24 MLT, with occurrence peaks existing in aurora regions around midnight and moving
toward the polar cap regions when approaching the sunset time. Values of the irregularity occurrence rate
are around 10-20% during the nighttime and can reach 30% for peak regions. The middle two panels
show the occurrence rate of failed inverted RO events. It is depicted that the peaks are located in aurora
regions and extent to the polar cap regions from sunset time to midnight. The increase in the occurrence
rate around these regions might be affected by the high occurrence rate of Es. The failed inverted RO
occurrence rates are also higher during the daytime in both hemispheres although the irregularity
occurrence rates are lower than 10% during this period. The right panels show the bending angle
oscillation results. Its value is larger in aurora regions around midnight, which can reach 2.1 μrad. For
those values in the South Hemisphere, strong bending angle oscillation can be seen in the polar cap
regions no matter during the daytime or nighttime. Besides, the peak regions locate mainly between
80° S-90° S instead of the 70° S-80° S as the irregularity occurrence peak shown in the bottom middle



panel.

We also use the scatter plot to study correlations between the ionospheric irregularity and the two
parameters. The results were plotted in Figure 4. Considering patterns of the failed inverted RO
occurrence rate and the bending angle oscillation could not agree very well with those of the irregularity
in high latitude regions, we mainly pay attention to the results in low and middle latitudes (60° S-60° N).
Correlations during the daytime (6~18 MLT) and the nighttime (0~6, 18~24 MLT) were displayed
respectively. Overall, the correlations between the ionospheric irregularity and the two parameters are
significant although they are not strictly liner especially for the bending angle oscillation during the
daytime. The scatters in the panels are probably due to that the bending angle oscillation is not only
affected by the Es and FI but also related to other factors. Meanwhile, the observation and inversion noise
could also make contributions.

In Figures 2-4, we mainly concern about the yearly average pattern. As stated above, the seasonal
variation of irregularity has been confirmed by previous studies (Arras et al., 2008; Chen and Huang,
2017). So the seasonal dependency of the failed inverted RO event and the bending angle oscillation
could also exist. For further investigating, the occurrence rate variation with Lon-Lat and MLT-MLat
were depicted in Figure 5 and Figure 6, respectively. Equinox (March, April, September, and October),
Northern Summer (May, June, July, and August), and Northern Winter (January, February, November,
and December) are considered here. Generally, in Figure 5, the distributions of the failed inverted RO
event occurrence are in good agreement with those of the irregularity. Both parameters are larger within
±30° in Equinox and larger in summer than in winter. For Northern Summer, the occurrence peaks are
near the North Africa area and the East Asia area. For Northern Winter, the peaks are available in the
Pacific Ocean regions nearby South America. Similar to the average pattern in Figure 2, the failed
inverted RO occurrence rate is high in polar regions even though the Es and FI occurrence is not obvious
in comparison with those of peak regions in low latitudes. The MLT-MLat distributions of the irregularity
and the failed inverted RO in Figure 6 also show similar seasonal variations. But for the southern polar
region in Equinox and the northern polar region in Northern Winter, the failed inverted RO occurs
significantly on condition that the occurrences of Es and FI are not obvious. This might be due to that



the FI occurrence is lower than those of Es in high latitudes but it plays the main role in the failed inverted
RO occurrence in these months. For the bending angle results in both bottom panels in Figures 5 and 6,
it is apparent that the bending angle oscillation value also follows a similar seasonal variation with the
ionospheric irregularity occurrence, which is larger in summer than in winter with the equinox as the
transitory season. It is noticeable that for the geographic distribution, larger values exist around Northern
Antarctica in all seasons. For the geomagnetic distributions, the three peaks along geomagnetic latitudes
are available in all seasons.

**6.   Conclusions and Implications**

In this paper, we focus on the ionospheric irregularity effects on GNSS atmospheric RO. The failed
inverted RO events and the bending angle oscillation are the two main parameters we concerned about.
The COSMIC S4 index provided by CDAAC during 2011-2013 is used to characterize the ionospheric
irregularity occurrence rate such as the Es and the F region irregularity. The 'bad' attribute in the atmPrf
file is used to identify the failed inverted RO events on condition that its value equals to 1. The mean
bending angle oscillation also from the atmPrf file is used to reflect the degree of bending angle
oscillation. Results from single cases are analyzed firstly. Then the distribution patterns and seasonal
variations of the ionospheric irregularity occurrence rate, failed inverted RO event occurrence rate, and
the bending angle oscillation are presented for the correlation study. The main conclusions and
implications of the paper are summarized in the following:
(1) The ionospheric irregularity such as the Es and the F region irregularity could affect the GNSS

311        atmospheric RO in terms of causing failed inverted RO events and the bending angle oscillation in

312        both cases and statistically.

(2) In middle and low latitudes, during the daytime, both the failed inverted RO event and the bending

314        angle oscillation are mainly affected by the Es. During the nighttime, the F region irregularity

315        contributes to the obvious increases of the failed inverted RO occurrence rate and the bending angle

316        oscillation around the geomagnetic equatorial regions.





(3) In the polar regions, the Es mainly affect the two parameters in the aurora regions from sunset to midnight. But the correlations between the ionospheric irregularity and the two parameters are not as obvious as those in middle and low latitudes.

(4) Seasonal dependency of the failed inverted RO occurrence and the bending angle oscillation exists, which also accord well with the seasonal variation of the Es and the F region irregularity.

(5) The occurrence rate of the failed inverted RO can reach 15% in low latitudes and even 20% in peak regions. It means that hundreds of COSMIC RO events per day will be ruled out during quality control. The bending angle oscillation between 60 and 80 km also varies from ~0.6 $\mu$rad in trough regions to ~2.5 $\mu$rad in peak regions. Although 60~80 km is not the main altitude range of RO data, the small-scale effects in atmospheric RO exist in all altitudes and could affect the atmospheric research related to RO products. Awareness of the ionospheric irregularity effect on RO could be beneficial to improve the data retrieval, quality control of GNSS atmospheric RO data processing and data assimilation application in numerical weather prediction (Cardinali and Healy, 2014).

**7. Acknowledgement**

This work was supported by the B-type Strategic Priority Program of the Chinese Academy of Sciences (Grant No. XDB41000000), the Open Research Project of Large Research Infrastructures - "Study on the interaction between low/mid-latitude atmosphere and ionosphere based on the Chinese Meridian Project", the National Natural Science Foundation of China (41427901), and the Key Research Program of the IGGCAS with Grant No. IGGCAS-201904. The University Corporation for Atmospheric Research (UCAR) COSMIC Data Analysis and Archive Center (CDAAC) is appreciated for processing and sharing the COSMIC radio occultation data to the community over years. All the data used in the study could be downloaded from the CDAAC website (http://cdaac-www.cosmic.ucar.edu/).

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

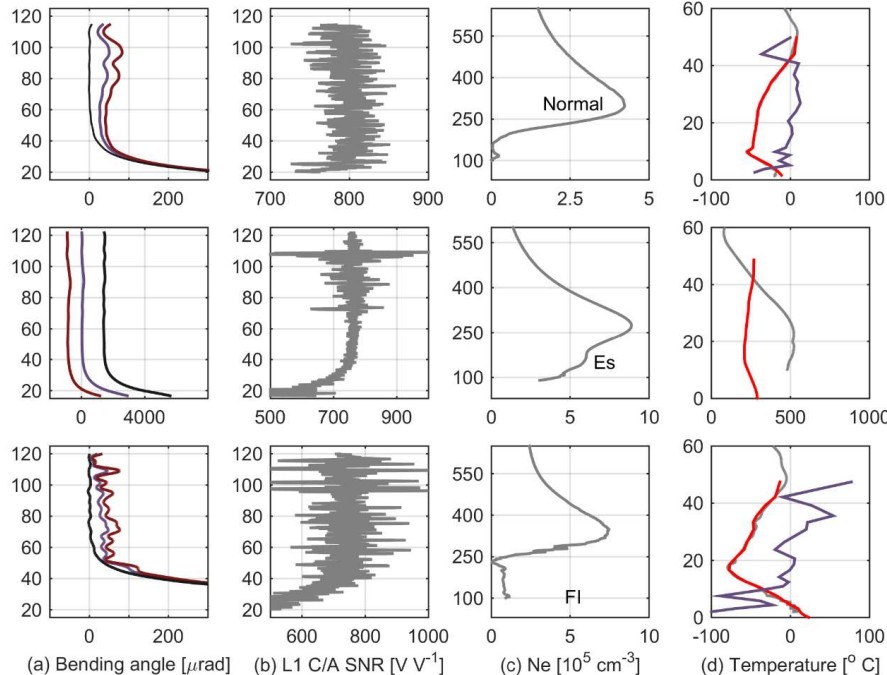

**Figure 1. Example of three cases occurred in 2013 made by COSMIC. Panels from top to bottom are normal example without the ionospheric irregularity occurrence, the Es example, and the F region irregularity example. Panels from left to right are the inverted L1 (purple line), L2 (brown line), and LC (black line) bending angles; the L1 C/A SNR, the electron density profile at the RO tangent points; the inverted dry temperature (grey line) versus the ECMWF results (red line) as well as their bias multiplied by 10 (purple line).**

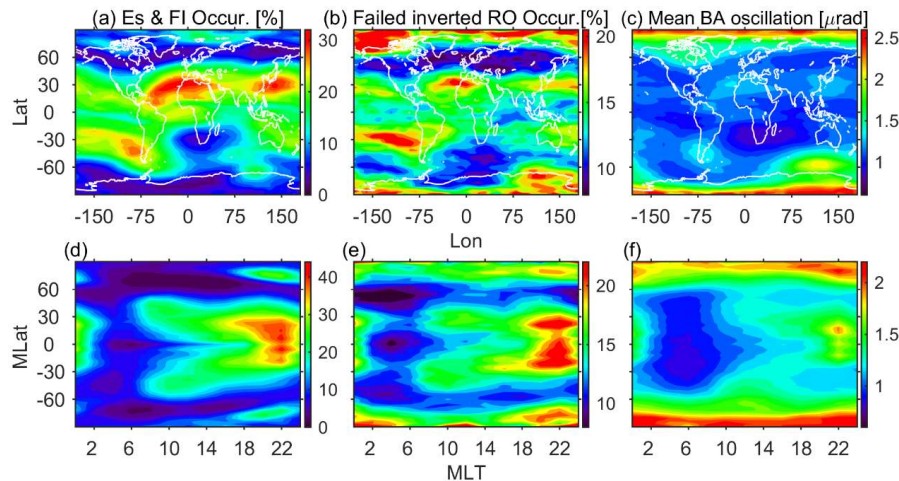


**Figure 2. Global geographical (top panels) and geomagnetic distributions (bottom panels) of the**

**Es and F layer irregularity occurrence rate (left panels), the failed inverted RO event occurrence**

**rate (middle panels), and the mean bending angle oscillation (right panels) during 2011-2013.**



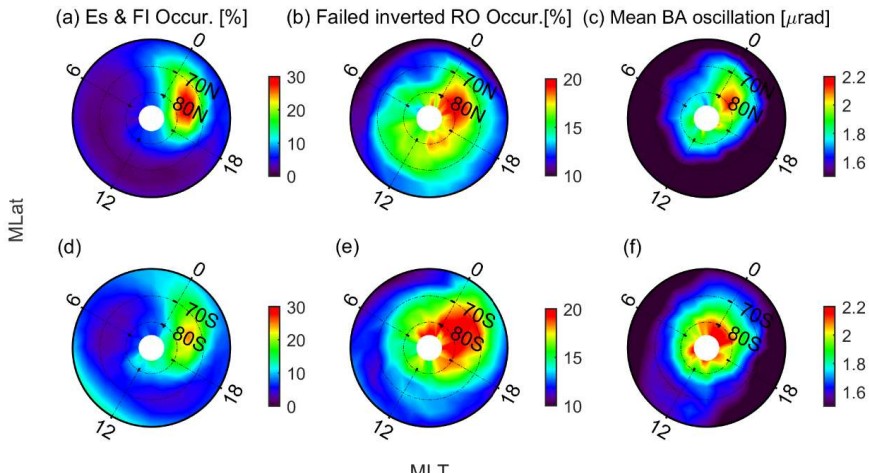


**Figure 3. MLT-MLat variation of the Es and F layer irregularity occurrence rate (left panels), the failed inverted RO occurrence rate (middle panels), and the mean bending angle oscillation (right panels) in polar regions. Please note that the top panels represent the results in northern polar regions while the bottom panels denote the southern polar regions.**







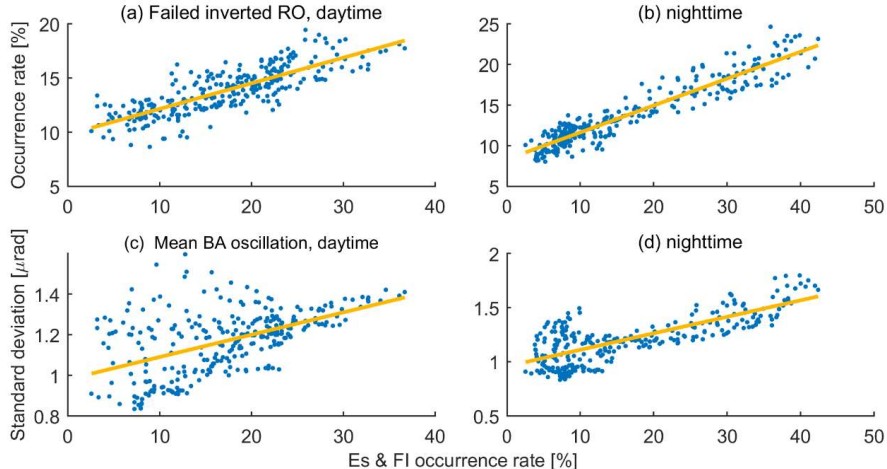


**Figure 4. Correlations between the ionospheric irregularity and the two parameters in middle and low latitudes (60° S-60° N) during the daytime (6~18 MLT, left panels) and nighttime (0~6 & 18~24 MLT, right panels) of 2011-2013. The yellow line is the corresponding linear least square fitting results.**


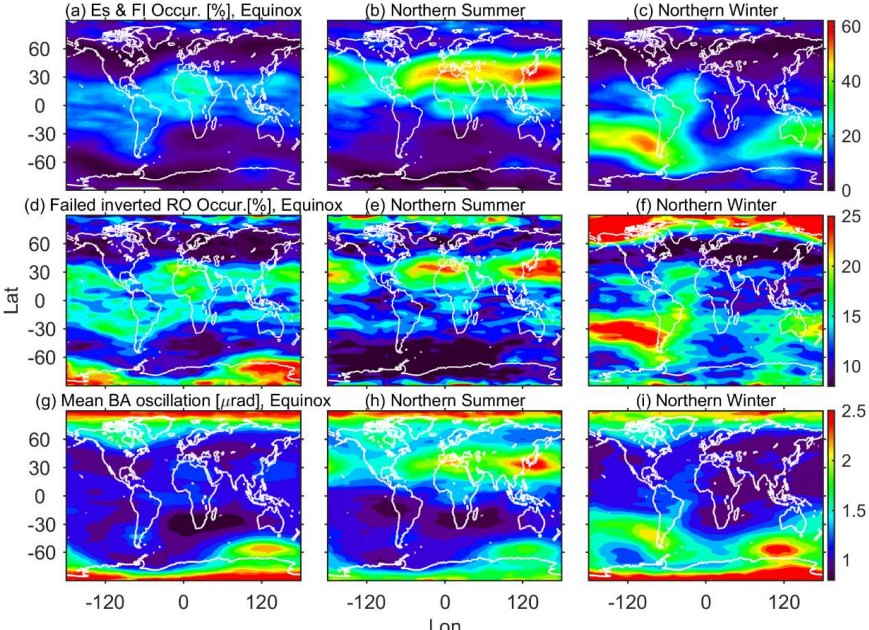


**Figure 5. Global geographical distribution of the Es and F layer irregularity occurrence rate (top panels), the failed inverted RO occurrence rate (middle panels), and the mean bending angle oscillation (bottom panels) for Equinox (left panels), Northern Summer (middle panels), and Northern Winter (right panels).**





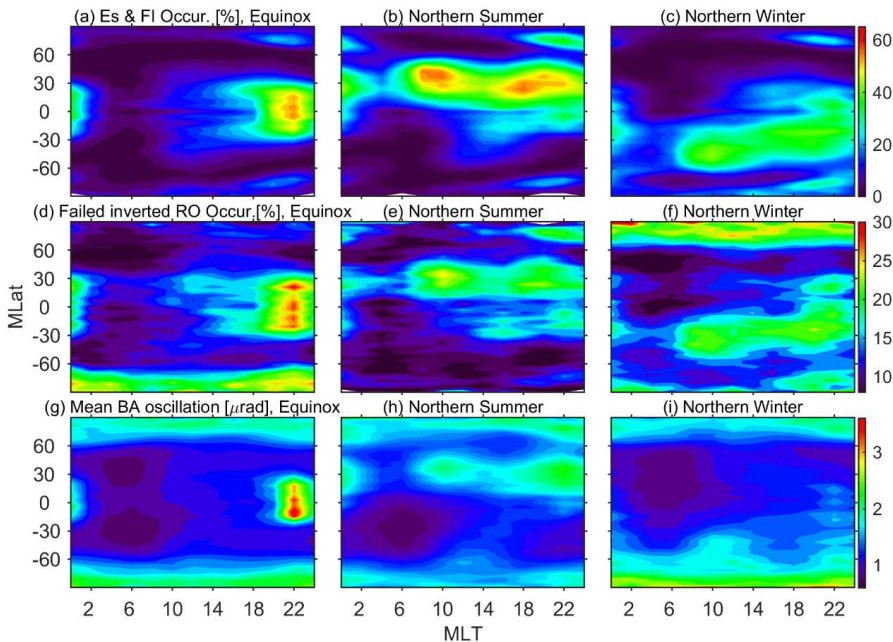

**Figure 6. The same as Figure 5, but for geomagnetic local time (MLT) and geomagnetic latitude (MLat) variation.**