# Peer review of "Statistical analyzing the effect of ionospheric irregularity"

_Atmospheric Measurement Techniques, 2020_

## Referee Comment (RC1) · Anonymous Referee #1 · 14 Jan 2021

**Statistical analyzing the effect of ionospheric irregularity on GNSS radio occultation atmospheric measurement**

Li and Yue

AMT-Review

General comment:

The submitted manuscript quantifies the impact of ionospheric irregularities on failed radio occultation (RO) inverted events and bending angle oscillations, due to the contribution of sporadic E occurrences and the F1-layer. The manuscript provides a very interesting contribution on assessing the impact of small-scall residual ionospheric errors (RIEs) on RO data profiles. Except of a few minor revisions, I recommend the manuscript for publication.

Specific comments:

Literature research (Introduction): Some of the older and newer published literature on assessing RIEs is missing. Regarding older literature the study of Danzer et al. 2013 is very interesting, since it applies a very similar approach to assess the impact of the RIE. The study analyzed the bending angle bias as a difference of the observed bending angle to the MSIS reference climatology, to obtain an estimate of the RIE. This similarity in the approaches should be mentioned. More recent literature is the work by Danzer et al. 2020 and Liu et al. 2020, where the former assesses the climatological impact of the RIE and the latter applies a profile-to-profile study which calculates the RIE through applying four terms (electron density, geomagnetic field, raypath inbound, and raypath outbound effects). Please have a look into it and discuss it in your manuscript. Maybe it could be also relevant to the calculation of small-scale scintillations (kappa and bi-local correction).

Line 114: Please provide information about the settings for the NCAR climatology model. How was the setting for the solar activity? Danzer et al. 2013 uses for example a constant solar flux value, in order to assess the solar variations of the RO bending angle to the climatology.

Line 118: Please provide a more detailed introduction of the S4 index, to aid readability and understanding.

Lines 193-196, Figure 2: I am curious, what is the statistics within a bin? I know it varies with latitude, however, just on average, from low to high latitudes, so that I get an idea of the numbers behind the percentage shown in Figure 2. Please also replot Figure 2, fix the range of the colorbar, e.g. column 2, the range moves from 0 to 20, and from 0 to something like 25. Furthermore, a colorbar needs a unit and a label of what is shown, the manuscript always includes this information in the title.

Lines 322-329: I really appreciated the summary, but I think the implication and discussion part could be extended. What are the next best steps to include the ruled-out events in the processing? Bending angle oscillations at high altitudes are usually handled via the high-altitude initialization, to remove such effects. Is the bi-local correction (Liu et al. 2020) an option for correcting such small-scall ionospheric scintillations?

Technical comments:

Figures: Please *improve readability of the Figures* 1,2,5,6 for the reader. For example, Figure 1 misses the information of label and unit on the y-axis (altitude, km). In the third column, the y-axis extent is not constant (it jumps a bit). Provide ticks in-between (third and fourth column, x- and y-axis). And so on … Figures 5 and 6: Include a unit and label information for the colorbar (r.h.s.), include a larger space between the colorbar and the plots.

---

## Referee Comment (RC2) · Anonymous Referee #2 · 26 Jan 2021

This manuscript investigated the effect of ionospheric irregularity on atmospheric RO data. Specifically, the authors took the S4 index as a proxy of the occurrence of ionospheric irregularity, and examined two atmospheric metrics - failure rate of atmospheric RO retrievals, and bending angle oscillation. Through comparing the morphology and seasonal dependency of two atmospheric metrics with the ones from irregularity occurrence, the authors demonstrated the pattern similarity between them, which indicated the impact of irregularity on the atmospheric retrievals. It is overall an interesting study and well written. The topic fits well to the scope of the AMT. Therefore, I would recommend this manuscript for publication after a few minor revisions.

[Figure]

Specific Comments: L144: LC –> ionospheric-corrected LC L217-218: The occurrence rates of Es and FI are mixed together in this study, which makes the interpretation of results a bit complicated, though the authors already tried their best to explain the impact of Es and FI separately in the text. What if you do the occurrence rate analysis separately for Es and FI based on the altitude of the S4 maximum? L288-292: Is there any proof to support the statement that the FI occurrence plays the main role in the failed inverted RO occurrence? Beside the large ionospheric residual caused by ionospheric irregularity, atmospheric RO data can be identified as "Bad" for many different reasons. For example, over the time period selected in this manuscript, 2011-2013, there are couple of sudden warming events happened, which makes the atmospheric structure changes significantly and far from the climatology, especially over the polar winter. The RO retrieval could be "Bad" because of the large deviation from the climatology under such condition. Fig. 2: What if the figure just focuses on the low and mid-latitude herein since you already presented the polar results in Fig. 3? Also, you might consider to use the discrete color bar instead of the gradient one. Fig. 3. This is a polar coordinate, not a Cartesian one. It's better to delete "MLAT" and "MLT" herein.
* * *

---

## Author Comment (AC1) · 20 Feb 2021

Response to Reviewer 1 Thanks for the invaluable comments and suggestions that helped to improve the quality of this manuscript. Considering all the comments, point-to-point responses are addressed in this rebuttal as follows.

General comment: The submitted manuscript quantifies the impact of ionospheric irregularities on failed radio occultation (RO) inverted events and bending angle oscillations, due to the contribution of sporadic E occurrences and the F1-layer. The manuscript provides a very interesting contribution on assessing the impact of small-scall residual ionospheric errors (RIEs) on RO data profiles. Except of a few minor

revisions, I recommend the manuscript for publication.

Reply: We have checked and revised the manuscript. Thank you very much for the helpful comments.

Specific comments: Literature research (Introduction): Some of the older and newer published literature on assessing RIEs is missing. Regarding older literature the study of Danzer et al. 2013 is very interesting, since it applies a very similar approach to assess the impact of the RIE. The study analyzed the bending angle bias as a difference of the observed bending angle to the MSIS reference climatology, to obtain an estimate of the RIE. This similarity in the approaches should be mentioned. More recent literature is the work by Danzer et al. 2020 and Liu et al. 2020, where the former assesses the climatological impact of the RIE and the latter applies a profile-to-profile study which calculates the RIE through applying four terms (electron density, geomagnetic field, raypath inbound, and raypath outbound effects). Please have a look into it and discuss it in your manuscript. Maybe it could be also relevant to the calculation of small-scale scintillations (kappa and bi-local correction).

Reply: Thanks for the comment. We have discussed the kappa and bi-local correction methods as "Danzer et al. (2013) have analyzed the bending angle bias of CHAMP and COSMIC RO data from 2001-2011 and tried to parameterize bending angle bias versus the solar cycle to make statistical corrections. Healy and Culverwell (2015) found a good correlation between the RIE and the difference of GPS L1 and L2 bending angles at the same impact parameter. They then proposed a correction method using the 'kappa' parameter under the ionospheric spherical symmetry assumption. This method was further tested by Danzer et al. (2015), Angling et al. (2018) and Danzer et al. (2020). It can reduce the systematic errors vary with solar cycle from 0.2 K to 2.0 K at altitudes between 40 km to 45 km. Liu et al. (2018) have analyzed the ionospheric structure influences on RIE in bending angles based on ray tracing simulations and further developed a "Bi-local correction approach" to calculate the RIE through an equation. This method considers both the ionospheric asymmetry effects
as well as the geomagnetic effects on bending angles (Liu et al., 2020)". The related references have also been added in the manuscript.

Line 114: Please provide information about the settings for the NCAR climatology model. How was the setting for the solar activity? Danzer et al. 2013 uses for example a constant solar flux value, in order to assess the solar variations of the RO bending angle to the climatology.

Reply: Thanks for the comment. During the atmospheric RO inversion, the NCAR climatology model is used to provide the neutral refractivity for calculating the model bending angle. Then the climatology model bending angle is used for statistical optimization of the observed bending angle. The bias between the observed bending angle and the climatology model bending angle can be used to evaluate the ionospheric effects on the observed bending angle, on condition that only the neutral atmospheric information is provided by the climatology model bending angle. It means no solar activity setting is needed for the NCAR climatology model. Danzer et al. (2013) use a constant solar flux value for the setting of the NeUoG ionosphere model. They use the operational analysis fields from ECMWF to provide the atmospheric information.

Line 118: Please provide a more detailed introduction of the S4 index, to aid readability and understanding.

Reply: Thanks for the comment. We have added the introduction of the S4 index as "Besides, the amplitude and phase measurements can be used to calculate the ionospheric scintillation index such as the S4 index. The S4 index is defined as the standard deviation of the received signal power normalized to the average signal power, it can represent the occurrence of the ionospheric irregularity (Yue et al., 2016)".

Lines 193-196, Figure 2: I am curious, what is the statistics within a bin? I know it varies with latitude, however, just on average, from low to high latitudes, so that I get an idea of the numbers behind the percentage shown in Figure 2. Please also replot Figure 2, fix the range of the colorbar, e.g. column 2, the range moves from 0 to 20,

and from 0 to something like 25. Furthermore, a colorbar needs a unit and a label of what is shown, the manuscript always includes this information in the title. Reply: Thanks for the comment. We use the geographical coordinate and the geomagnetic coordinate of the tangent point to decide whether a RO event belongs to a bin. For the ionospheric irregularity in a bin, the occurrence rate is calculated by

"Occurrence rate =" "number of RO events (S4>0.3)" /"number of RO events" $\times 100\%$ (1)

It means the occurrence rate of a bin is the ratio of the number of RO events with S4 index > 0.3 and the number of all RO events in this bin. Similarly, for the failed inverted RO event, the occurrence rate is calculated by

Occurrence rate =(number of failed inverted RO ("bad RO" ))/(number of RO events)$\times 100\%$ (2)

The median value of the smean parameter of all RO events in a bin is calculated to represent the bending angle oscillation in this bin. The range of colorbars have been revised, the label and unit have also been added to the colorbars. Please note that we have added a figure after Figure 1 as the new Figure 2. It shows the occurrence rates of Es and FI separately. As a result, the serial numbers of figures after Figure 1 have been changed in orders.

Lines 322-329: I really appreciated the summary, but I think the implication and discussion part could be extended. What are the next best steps to include the ruled-out events in the processing? Bending angle oscillations at high altitudes are usually handled via the high-altitude initialization, to remove such effects. Is the bi-local correction (Liu et al. 2020) an option for correcting such small-scall ionospheric scintillations?

Reply: Thanks for the comment. We think a suitable filter could be helpful in reducing the ionospheric irregularity effects on bending angles and including ruled-out events in the processing. The potential method needs to be more investigated and we hope to

study it in our further work. Besides, the causing of failed inverted RO events in high latitudes is also needed to be studied. As depicted in the new Figure 5, the correlations between the failed inverted RO events and the ionospheric irregularity are not as good as those in low and middle latitudes. We have added a small discussion in the conclusion part as "Overall, the ionospheric irregularity effects on GNSS atmospheric RO measurement exist in terms of failed RO event inversion and bending angle oscillation statistically. It makes the calibration of ionospheric irregularity effects on RO more challenging and urgent. A suitable filter may be helpful in reducing the ionospheric irregularity effects on bending angles and including ruled-out events in the processing. The potential method needs to be more investigated and we hope to study it in our further work. Besides, the causing of failed inverted RO events in high latitudes is also needed to be studied". According to Mannucci et al. (2013) and Liu et al. (2020) have discussed, the "bi-local" correction is mainly used to correct the large-scale RIE caused by the separation of GNSS signals with different frequencies and the RIE caused by the geomagnetic effects. The correction of small-scale ionospheric scintillations still needs to be further studied.

Technical comments: Figures: Please improve readability of the Figures 1,2,5,6 for the reader. For example, Figure 1 misses the information of label and unit on the y-axis (altitude, km). In the third column, the y-axis extent is not constant (it jumps a bit). Provide ticks in-between (third and fourth column, x- and y-axis). And so on . . . Figures 5 and 6: Include a unit and label information for the colorbar (r.h.s.), include a larger space between the colorbar and the plots.

Reply: Thanks for the comment. The missed information in Figure 1 has been added. The unit and label information for the colorbar have been added to the new Figure 3, 4, 6, 7. The space between the colorbar and plot has also been adjusted.

References Danzer, J., Scherllin-Pirscher, B., and Foelsche, U.: Systematic residual ionospheric errors in radio occultation data and a potential way to minimize them, Atmospheric Measurement Techniques, 6, 2169-2179, doi.org/10.5194/amt-6-2169-

2013, 2013. Liu, C., Kirchengast, G., Syndergaard, S., Schwaerz, M., Danzer, J., and Sun, Y.: New Higher-Order Correction of GNSS RO Bending Angles Accounting for Ionospheric Asymmetry: Evaluation of Performance and Added Value, Remote Sensing, 12, doi.org/10.3390/rs12213637, 2020. Mannucci, A. J., Ao, C. O., Pi, X., and Iijima, B. A.: The impact of large scale ionospheric structure on radio occultation retrievals, Atmospheric Measurement Techniques, 4, 2837-2850, doi.org/10.5194/amt-4-2837-2011, 2011.

Fig. 1.

[Figure]

**Fig. 2.**

[Figure]

**Fig. 2.**

[Figure]

**Fig. 3.**

[Figure]

Fig. 4.

[Figure]

**Fig. 5.**

(a) Es & FI, Equinox
(b) Northern Summer
(c) Northern Winter

Occurrence rate (%)

(d) Failed inverted RO, Equinox
(e) Northern Summer
(f) Northern Winter

Occurrence rate (%)

(g) Mean BA oscillation, Equinox
(h) Northern Summer
(i) Northern Winter

Bending angle ($\mu$rad)

Lat

Lon

**Fig. 6.**

[Figure]

Fig. 7.

---

## Author Comment (AC2) · 20 Feb 2021

Response to Reviewer 2 Thanks for the invaluable comments and suggestions that helped to improve the quality of this manuscript. Considering all the comments, point-to-point responses are addressed in this rebuttal as follows.

General comment: This manuscript investigated the effect of ionospheric irregularity on atmospheric RO data. Specifically, the authors took the S4 index as a proxy of the occurrence of ionospheric irregularity, and examined two atmospheric metrics - failure rate of atmospheric RO retrievals, and bending angle oscillation. Through comparing the morphology and seasonal dependency of two atmospheric metrics with the ones

from irregularity occurrence, the authors demonstrated the pattern similarity between them, which indicated the impact of irregularity on the atmospheric retrievals. It is overall an interesting study and well written. The topic fits well to the scope of the AMT. Therefore, I would recommend this manuscript for publication after a few minor revisions.

Reply: We have checked and revised the manuscript. Thank you very much for the helpful comments.

Specific Comments: L144: LC –> ionospheric-corrected LC

Reply: Thanks for the comment. It has been revised as "ionospheric-corrected LC".

L217-218: The occurrence rates of Es and FI are mixed together in this study, which makes the interpretation of results a bit complicated, though the authors already tried their best to explain the impact of Es and FI separately in the text. What if you do the occurrence rate analysis separately for Es and FI based on the altitude of the S4 maximum?

Reply: Thanks for the comment. On condition that the morphologies of the failed inverted RO events and the bending angle oscillation are related to both Es and FI, we plot the occurrence rate of Es and FI together for discussion. For convenient comparison, a figure has been added as the new Figure 2 to display the occurrence rates of Es and FI separately. Please note that the serial numbers of figures after Figure 1 have been changed in orders.

L288-292: Is there any proof to support the statement that the FI occurrence plays the main role in the failed inverted RO occurrence?

Reply: Thanks for the comment. It is a guess made by us. We are sorry for our neglect here. The related description has been removed. We hope to study the causing of failed inverted RO events and bending angle oscillation in high latitudes more specifically in our further study.

Beside the large ionospheric residual caused by ionospheric irregularity, atmospheric RO data can be identified as "Bad" for many different reasons. For example, over the time period selected in this manuscript, 2011-2013, there are couple of sudden warming events happened, which makes the atmospheric structure changes significantly and far from the climatology, especially over the polar winter. The RO retrieval could be "Bad" because of the large deviation from the climatology under such condition.

Reply: Thank you very much for the comment. In this paper we mainly concentrate on studying the correlation between the failed inverted RO events and the ionospheric small-scale irregularity. As your suggestion, the sudden warming events could be an important reason for the failed inverted RO events. It is essential for us to investigate its effect on the RO bending angle.

Fig. 2: What if the figure just focuses on the low and mid-latitude herein since you already presented the polar results in Fig. 3? Also, you might consider to use the discrete color bar instead of the gradient one. Fig. 3. This is a polar coordinate, not a Cartesian one. It's better to delete "MLAT" and "MLT" herein.

Reply: Thanks for the comment. As depicted in the new Figure 5, the correlations between the failed inverted RO events and the ionospheric irregularity, are not as good as those in low and middle latitudes. That's the reason why we mainly concentrate on the study in low and middle latitudes in the following part in this manuscript. We hope to study the causing of failed inverted RO events and bending angle oscillation in high latitudes more specifically in our further work. Both the ionospheric irregularity effects and the sudden warming event effects should be taken into consideration. The colorbars of all figures have been revised and the "MLAT" and "MLT" in the new Figure 4 has been deleted.

[Figure]

Fig. 1.

Altitude (km)

Normal

Es

Fl

(a) Bending angle [$\mu$rad]  (b) L1 C/A SNR [V V$^{-1}$]  (c) Ne [$10^5$ cm$^{-3}$]  (d) Temperature [$^o$ C]

**(a) Es Occurrence Rate**

**(b) FI Occurrence Rate**

**(c) Es Occurrence Rate**

**(d) FI Occurrence Rate**

**Fig. 2.**

[Figure]

**Fig. 3.**

[Figure]

**Fig. 4.**

[Figure]

**Fig. 5.**

[Figure]

**Fig. 6.**

[Figure]

Fig. 7.

---

## Author Response (AR2)

**Atmospheric Measurement Techniques**

Manuscript: amt-2020-440
Statistical analyzing the effect of ionospheric irregularity on GNSS radio occultation atmospheric measurement, by Mingzhe Li and Xinan Yue

Dear editor:

   Thank you very much for your handing of our paper. We have responded to the comments point by point and made revisions to the manuscript accordingly. Here we submit the response to the comments and a revised manuscript. Our point by point replies are given below. The comments are marked in italic, and revised portions are marked in blue. The line numbers of revisions are cited according to the revised manuscript.

Sincerely
Mingzhe Li

**Comments to the Author:**

*As mentioned by Reviewer 2 you should at least mention that there are several possible sources for "bad" data, for example the sudden stratospheric warmings.*

**Reply:** Thanks for the comment. We have mentioned the sudden stratospheric warming events effect in lines 284-291 as "The scatters in the panels are probably due to that the failed inverted RO and the bending angle oscillation are not only affected by the Es and FI but also related to other factors. For example, the sudden stratospheric warming events can make the atmospheric structure changes significantly and far from the climatology. As a result, bias between the RO observation and the climatological model could be increased and lead to the atmosphere RO event is identified as "Bad" during inversion. Considering the sudden stratospheric warming events often occur over the polar winter (Butler et al., 2015), they could also contribute to the pattern difference between the ionospheric irregularity occurrence rate and the two parameters in Figure 4. Meanwhile, the observation and inversion noise could also make contributions".

*Also, you must improve grammar in the new lines 347-352.*

**Reply:** Thanks for the comment. It has been revised in lines 347-350 as "Overall, the ionospheric irregularity effects on GNSS atmospheric RO measurement exist. The effects can lead to the failed inverted RO event and the bending angle oscillation. A suitable filter may be effective in calibrating these effects and improving the quality of atmospheric RO products. We hope to investigate the potential calibrating method in our further work".

We have also checked the manuscript and revised some spelling errors. Thank you very much for the helpful suggestions.